# Determinants of socioeconomic factors for quality of life and depressive symptoms in community-dwelling older people: A cross-sectional study in Brazil and Portugal

Larissa Silva Sadovski Torres[1], Adriana Catarina de Souza Oliveira[2°], Mayara Priscilla Dantas Araújo[1°], Maria Débora Silva de Carvalho[3°], Lívia Batista da Silva Fernandes Barbosa[3°], Bruno Araújo da Silva Dantas[4°]*, Carmelo Sergio Gómez Martínez[2°], Francisco Arnoldo Nunes de Miranda[5°], Felismina Rosa Parreira Mendes[6°], Gilson de Vasconcelos Torres[7°]

1 Health Sciences Center, Federal University of Rio Grande do Norte, Natal, State of Rio Grande do Norte, Brazil, 2 Faculty of Nursing, Catholic University of Murcia, Murcia, Spain, 3 Department of Nursing, Federal University of Rio Grande do Norte, Natal, State of Rio Grande do Norte, Brazil, 4 School of Health Sciences of Trairi, Federal University of Rio Grande do Norte, Santa Cruz, State of Rio Grande do Norte, Brazil, 5 Department of Nursing and Graduate Program in Nursing, Federal University of Rio Grande do Norte, Natal, State of Rio Grande do Norte, Brazil, 6 São João de Deus School of Nursing, University of Évora, Évora, Portugal, 7 CNPQ Researcher (PQ1D) and Department of Nursing, Federal University of Rio Grande do Norte, Natal, State of Rio Grande do Norte, Brazil

° These authors contributed equally to this work.
* bruno.dantas@ufrn.br

## Abstract

Our aim was to analyze the association between socioeconomic status and quality of life (QoL) among older people with depressive symptoms treated through the Primary Health Care (PHC) system in Brazil and Portugal. This was a comparative cross-sectional study with a nonprobability sample of older people in the PHC in Brazil and Portugal conducted between 2017 and 2018. To evaluate the variables of interest, the socioeconomic data questionnaire, the Geriatric Depression Scale and the Medical Outcomes Short-Form Health Survey were used. Descriptive and multivariate analyses were performed to test the study hypothesis. The sample consisted of n = 150 participants (Brazil n = 100 and Portugal n = 50). There was a predominance of woman (76.0%, p = 0.224) and individuals between 65 and 80 years (88.0%, p = 0.594). The multivariate association analysis showed that in the presence of depressive symptoms, the QoL mental health domain was most associated with the socioeconomic variables. Among the prominent variables, woman group (p = 0.027), age group 65–80 years (p = 0.042), marital status "without a partner" (p = 0.029), education up to 5 years (p = 0.011) and earning up to 1 minimum wage (p = 0.037) exhibited higher scores among brazilian participants. The portuguese participants showed an association between the general health status domain and woman group (p = 0.042) and education up to 5 years (p = 0.045). The physical functioning domain was associated with income of up to 1 minimum wage (p = 0.037). In these domains, the portuguese participants exhibited higher scores than the brazilian participants. We verified the association between socioeconomic profile and QoL in the presence of depressive symptoms, which occurred mainly

**Data Availability Statement:** The data underlying the results presented in the study are available from Mendeley Data repository and are available at the link https://data.mendeley.com/datasets/r4h9gvrrw2.

**Funding:** Initials of the authors who received each award: GVT Grant numbers awarded to each author: 01 The full name of each funder: Conselho Nacional de Desenvolvimento Científico e Tecnológico URL of each funder website: https://www.gov.br/cnpq/pt-br The funders had no role in study design, data collection and analysis, decision to publish, or preparation of the manuscript.

**Competing interests:** The authors have declared that no competing interests exist.

among woman, participants with low levels of education and low income, with QoL aspects related to mental, physical and social health and self-perceived health. The group from Brazil had higher QoL scores than the group from Portugal.

## Introduction

Depression is one of the most common mental disorders in the world population and can manifest as symptoms such as loss of appetite, sleep problems, concentration deficits and psychomotor agitation [1]. It is estimated that this disorder affects more than 264 million people worldwide, configuring itself as a disabling problem with great potential to change people's daily lives [2]. The World Health Organization (WHO) indicates that depression represents 4.3% of the global burden of morbidity and that a person with depression has up to a 60.0% greater chance of premature death compared to the general population [3].

Among the populations most affected, the older people stand out because, regardless of the prevalence of other chronic diseases in this group, physiological changes common to aging are observed as certain physical and functional limitations that, in turn, can significantly affect their health, mental health and quality of life (QoL) [4].

QoL, in turn, can be defined by the perception of the individual about his or her roles and relationships, considering physical, functional, and emotional health. Among its aspects, social interaction is represented by the individual's satisfaction with their relationships established with the people around them, including family, friends, co-workers and other activities [5]. The influence of QoL on mental disorders and, specifically, on depressive symptoms has been described in the literature as a major concern for public health, especially older people individuals [6].

Regarding this influence, it is important to highlight the concept of active aging presented by the WHO and described as the process of optimizing opportunities for improving health, social participation and security to preserve and improve QoL in the aging process. Ensuring that this happens also requires government policies aimed at social determinants and that health services establish an effective link with the community and its reality for the implementation of assertive measures to prevent injuries and promote health [7].

From the perspective of strategies for the prevention of diseases and the promotion of active aging with good QoL, Primary Health Care (PHC) is considered one of the main tools to meet the demands of population health [8]. However, the PHC network specializing in mental health care has been insufficient worldwide, especially for the older population. In this context, it is important to note that mental disorders such as depression and its symptoms may have different proportions according to the sociocultural and political aspects of each location [3].

Therefore, it is important to observe and compare data from countries with different contexts to understand these influences. In Brazil, the 2019 National Health Survey showed that depression was observed mainly in women, people with low income and those with little education. In addition, 13.0% of depression diagnoses in the country were in younger older people [9]. Portugal, which is a more developed country than Brazil, also presents great challenges regarding care for depression and its symptoms in older individuals. The National Health Survey, conducted in 2019, showed that 32.0% of portuguese respondents who had depressive symptoms were older people [10].

The present study was based on the understanding that, despite the existence of epidemiological data from both countries, there is a need for additional studies regarding the impact of socioeconomic factors on depressive symptoms and their relationship with QoL among older

adults. Thus, the United Nations launched the Decade of Healthy Aging in the Americas (2021 to 2030) and called for a collaboration to develop studies that investigate the main factors affecting QoL among older people [11]. Likewise, the WHO Comprehensive Mental Health Action Plan 2013–2030 established objectives, including the strengthening of data and scientific research in the field of mental health [3].

This study aims to analyze the association between socioeconomic status and QoL of older people with depressive symptoms treated at PHC centers in Brazil and Portugal. Our study hypothesis is that socioeconomic characteristics are determinants of the QoL of older people treated through PHC for depressive symptoms.

## Materials and methods

### Ethical considerations

The study was approved in Brazil by the Research Ethics Committee of the University Hospital Onofre Lopes (registration no. 562,318), in Portugal by the Research Ethics Committee of the University of Évora (registration no. 14011) and by the Ethics Committee for Scientific Research in the Fields of Health and Human Welfare at the University of Évora (registration no. 17.006/2018).

Prior to the interviews, participants were provided with guidelines and general clarifications regarding questions, risks and benefits before signing the Free and Informed Consent Form. Consent was obtained with the written signature of the term; one copy being given to the participant and the other kept by the researcher. The precepts of ethics, good clinical practice and volunteer status were respected, according to the Declaration of Helsinki.

### Study design and location

This is a cross-sectional, comparative study with a quantitative approach conducted in Brazil and Portugal from December 2017 to July 2018. In Brazil, the research was developed in the PHC units of the municipalities of Natal and Santa Cruz in the state of Rio Grande do Norte. In Portugal, the study took place at the PHC in the city of Évora.

### Population and sample

The study group comprised older people cared for at the PHC of each study setting. A non-probability and convenience sample was selected from participants who received regular care at the PHC (at least once a month) and who met the study inclusion criteria. According to the number of older people enrolled in the PHC units studied, the population of the Brazilian sample was estimated at n = 135 and for Portugal, n = 70 (confidence level of 95.0% and margin of error of 5.0%). The sample size calculation was performed, which resulted in a sample size of n = 100 (Brazil) and n = 60 participants (Portugal), for a total n = 160. The result was obtained with the aid of an online calculator, available at https://calculareconverter.com.br/calculo-amostral/. A total of 150 participants completed the study, 100 from Brazil and 50 from Portugal.

The criteria for inclusion in the study were as follows: being 65 years of age or older, the age established by the WHO for individuals to be considered older persons, being registered with PHC in their research setting (Brazil or Portugal) for at least six months before the study, and presenting unimpaired cognitive status that allowed participants to understand all the instruments; this criterion was evaluated with the Mini-Mental State Examination (MMSE), and required participants to obtain 17 points or more on the scale for inclusion in the study [12, 13].

The exclusion criteria were having permanent or transient physical disability at the time of data collection and personal or family trauma reported by the participant in a period less than or equal to six months prior to the time of the study.

## Instruments and variables

The instruments used were established using the validation criteria and considering the linguistic aspects between both portuguese-speaking countries.

To construct the profiles of the participants, we used the socioeconomic data questionnaire, containing questions about age group, gender, family income, education and marital status. These questions were transformed into variables categorized dichotomously to facilitate data treatment and eliminate possible bias in their analyses resulting from occasional and excessive fragmentation.

To measure depressive symptoms, we used the Geriatric Depression Scale (GDS-15), which asks about topics such as sadness, loneliness, suicidal ideation, mood swings, fear and difficulty in relating to other people. Responses regarding depressive symptoms are categorized as "absent", "mild", "moderate" and "severe" [14, 15]. To contemplate the objectives of the study, the symptoms were recategorized as "absent" and "present", and the latter category included the combination of the variables "mild", "moderate" and "severe".

To measure QoL, the portuguese version of the *Medical Outcomes Short-Form Health Survey* (SF-36) was used; the survey contains 36 questions that measure aspects such as physical, functional, emotional, mental health and self-perception of general health. Responses are measured on a Likert Scale that generates a score from 0 to 100. The closer the score is to zero, the worse the QoL [5].

## Data collection

Before performing the data collection, the research coordinators provided training for all interviewers regarding the application of the instruments selected for the study. They were administered by undergraduate and graduate students of the Federal University of Rio Grande do Norte and the University of Évora. Data collection occurred between december 2017 and july 2018, with an average duration of one hour for face-to-face interview, conducted by the researcher in an interview format with questions guided by the instruments. For this purpose, prior appointments were made with the participants, who met with researchers at their PHC units. Data collection was supervised by the coordinators of the research project.

Brazilians were chronologically the first to be surveyed. This group was used as a comparative baseline for participants from Portugal. To eliminate confounders, the socioeconomic profiles of brazilians were grouped according to the following variables: gender (woman or man), age group (65–80 or 81–100 years), marital status (with or without a partner), family income per minimum wage ($< 1$ or $> 1$) and education (up to 5 years and more than 5 years). Combinations of variables were given a code representative of socioeconomic status. Thus, each participant with the same combination of variables was assigned to the same code. The goal was to identify at least one individual from each country for each code. Therefore, codes that were not represented by at least two participants (one from Brazil and the other from Portugal) were discarded. There was no blinding of any researcher or research participant, and no one received any stimulus or remuneration.

## Data analysis and treatment

The Statistical Package for the Social Sciences (SPSS) version 20.0 software (International Business Machines Corporation [IBM], Armonk, NY, USA) enabled all statistical analyses, and the

data were tabulated and presented in tables with the aid of Microsoft® software's Excel 2016 (Microsoft Corporation, Washington, WA, USA).

The Shapiro-Wilk test found nonnormality of the variables of QoL and depressive symptoms. Therefore, Pearson's descriptive and nonparametric chi-square test was used for the socioeconomic data to analyze the level of equality between the two research groups and to analyze the association of the categorical variables of the GDS-15 (absent/present). We also measured the odds ratio (OR) for the categorical variables of the GDS-15 scale. For this analysis, we adopted the parameter > 1.0 for positive OR. The Mann-Whitney U test was used for the analysis of associations of the SF-36 scalar variables (expressed as 25th, 50th and 75th percentiles), as well as for the multivariate analysis between socioeconomic variables and QoL (SF-36) between Brazil and Portugal. The confidence interval (CI) adopted was 95.0%, and to determine the level of statistical significance, we considered the value of $p < 0.05$ [16]. The data were deposited in the *Mendeley Data* repository and are available at the link https://data.mendeley.com/datasets/r4h9gvrrw2.

## Results

The final sample was 150 participants, 100 brazilians and 50 portuguese. Some individuals were excluded in the group from Brazil. In Portugal, none were excluded. The sample selection process is shown in Fig 1.

Table 1 shows similarity in some variables of the socioeconomic profile of both groups, with a predominance of woman in the total sample (76.0%,p = 0.224) and individuals aged between 65 and 80 years (88.0%,p = 0.594). In Brazil, notable differences were a lower level of education (79.0%,p = 0.001) and the majority of participants did not live alone (86.0%, p = 0.001). In Portugal, all individuals had an income of less than one minimum wage (100.0%,p <0.001).

Table 2 presents the descriptive analysis of depressive symptoms (GDS-15). A predominance of symptoms was observed in the Brazilian group (59.0%) (p = 0.015,OR = 1.81–95%, CI = 1.12–2.81).

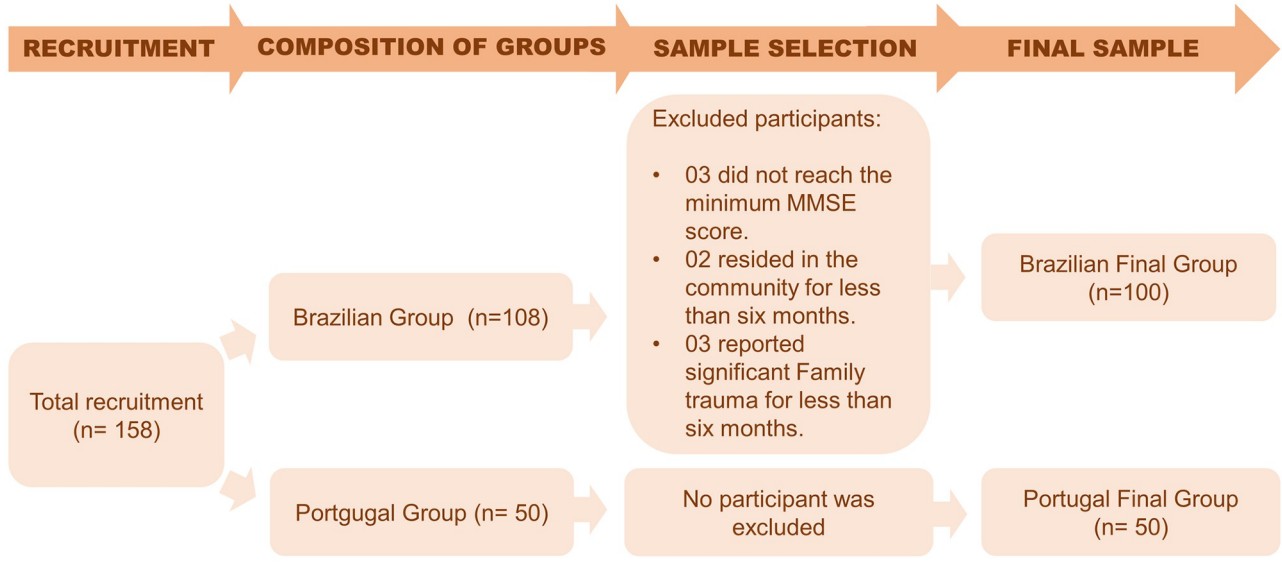

**Fig 1. Flowchart of the sample recruitment and selection process in Brazil and Portugal.**

**Table 1. Socioeconomic characteristics of participants from Brazil and Portugal.**

| Socioeconomic Aspects | | Brazil (n = 100) | Portugal (n = 50) | Total (n = 150) | p [a] |
|---|---|---|---|---|---|
| | | n (%) | n (%) | n (%) | |
| Gender | Woman | 73 (73.0) | 41 (25.6) | 114 (76.0) | 0.224 |
| | Man | 27 (27.0) | 9 (18.0) | 36 (24.0) | |
| Age range, yr | 65–80 | 89 (89.0) | 43 (86.0) | 132 (88.0) | 0.594 |
| | 81–100 | 11 (11.0) | 7 (14.0) | 18 (12.0) | |
| Marital status | Married/cohabitating | 49 (49.0) | 28 (56.0) | 77 (51.3) | 0.419 |
| | Single/widowed/divorced | 51 (51.0) | 22 (44.0) | 73 (48.7) | |
| Educational entertainment, yr | ≤ 5 | 79 (79.0) | 26 (52.0) | 105 (70.0) | 0.001 |
| | > 5 | 21 (21.0) | 24 (48.0) | 45 (30.0) | |
| Lives alone | Yes | 14 (14.0) | 19 (38.0) | 33 (22.0) | 0.001 |
| | No | 86 (86.0) | 31 (62.0) | 117 (78.0) | |
| Household income, [b] minimum wage | ≤ 1 | 42 (42.0) | 50 (100.0) | 92 (61.3) | <0.001[c] |
| | > 1 | 58 (58.0) | 0 (0.0) | 58 (38.7) | |

[a] Pearson's chi-squared test;

[b] Minimum wage, BRL 954.00 (BRL) in Brazil/€618.00 (EUR) in Portugal (2018);

[c] Fisher's exact test.

**Table 2. Characterization of depressive symptoms (GDS-15) in Brazil and Portugal.**

| Depressive Symptoms (GDS-15) | Brazil (n = 100) | Portugal (n = 50) | Total (n = 150) | p [a] OR [b] (CI 95%) |
|---|---|---|---|---|
| | n (%) | n (%) | n (%) | |
| Absentee | 41 (41.0) | 31 (62.0) | 72 (48.0) | 0.015 |
| Present | 59 (59.0) | 19 (38.0) | 78 (52.0) | 1.81 (1.12–2.81) |

[a] Pearson's chi-squared test;

[b] OR (odds ratio).

Table 3 presents the analysis of the distribution of scores by percentiles of the aspects of QoL. In the Brazilian group, the social role functioning domain (p = 0.022) and Total Score (p = 0.006) stood out. The group from Portugal exhibited higher scores in the physical functioning (p <0.001), general health perceptions (p = 0.043) domains and physical health dimension (p <0.001).

Table 4 shows the association between the categorical variables of depressive symptoms (absent/present) and the QoL scores. In Brazil, the absence of symptoms was associated with higher scores in the physical role functioning (p = 0.004), physical functioning (p = 0.016), total score (p = 0.002) and physical health dimension (p = 0.006). Portugal exhibited the same behavior in the domains of emotional role functioning (p = 0.007), physical functioning (p = 0.013), physical role functioning (p<0.001), vitality (p = 0.013), mental health (p = 0.005), social role functioning (p = 0.049), total score (p = 0.049) and in the dimensions mental health (p = 0.022) and physical health (p = 0.011). The pain domain exhibited higher scores in the presence of depressive symptoms (p = 0.002).

When performing the multivariate analysis of association between the socioeconomic variables, presence of depressive symptoms and QoL, we selected the most relevant results and present them in Table 5. The mental health domain was the domain most significantly

**Table 3. Characterization of quality of life scores in Brazil and Portugal.**

| QoL (SF-36) | Study Place | | | | | | |
|---|---|---|---|---|---|---|---|
| | Brazil (*n* = 100) | | | Portugal (*n* = 50) | | | |
| | Percentile | Mean (SD*) | CI* 95% | Percentile | Mean (SD) | CI 95% | p [a] |
| | 25-50-75 | | | 25-50-75 | | | |
| Domains | | | | | | | |
| Emotional role functioning | 33-100-100 | 70.3 (42.6) | 61.9–72.6 | 66-100-100 | 78.0 (39.0) | 66.9–89.0 | 0.359 |
| Physical functioning | 0-50-100 | 51.7 (44.2) | 43.0–60.5 | 75-100-100 | 82.0 (32.7) | 72.7–91.3 | <0.001 |
| Physical role functioning | 35-67-90 | 60.8 (30.1) | 54.8–66.8 | 50-75-90 | 67.9 (28.0) | 59.9–75.9 | 0.174 |
| Vitality | 44-55-60 | 51.8 (14.6) | 48.9–54.8 | 53-60-56 | 49.7 (12.0) | 46.3–53.1 | 0.348 |
| General health perceptions | 47-57-55 | 41.9 (18.0) | 38.3–45.5 | 49-58-56 | 47.2 (16.9) | 42.4–52.0 | 0.043 |
| Mental health | 45-50-60 | 56.3 (9.9) | 54.3–72.6 | 43-50-64 | 54.6 (11.2) | 51.5–57.8 | 0.332 |
| Pain | 40-50-60 | 38.9 (24.9) | 34.0–43.8 | 49-57-40 | 31.2 (21.4) | 25.1–37.3 | 0.076 |
| Social role functioning | 20-40-50 | 51.0 (15.7) | 47.9–54.1 | 20-30-50 | 46.5(11.0) | 43.4–49.6 | 0.022 |
| Total Score | 50-50-61 | 53.0 (11.6) | 50.7–55.3 | 37-50-63 | 57.57(10.7) | 54.5–60.6 | 0.006 |
| Dimensions | | | | | | | |
| Mental health | 52-56-61 | 54.2 (9.8) | 52.3–56.2 | 48-56-61 | 56.3(11.3) | 53.0–59.5 | 0.235 |
| Physical health | 30-35-57 | 48.7(11.9) | 46.4–51.1 | 35-50-63 | 55.6(9.2) | 53.0–58.2 | <0.001 |

[a] Nonparametric Mann-Whitney U test (p-value)

associated with socioeconomic variables. Among them woman group (p = 0.027), age group 65–80 years (p = 0.042), marital status "without a partner" (p = 0.029), education up to 5 years (p = 0.011) and income of up to 1 minimum wage (p = 0.037). In all these variables, higher scores were observed in the group from Brazil.

With the highest scores in Portugal, the domain general health perceptions was associated with woman group (p = 0.042) and education up to 5 years (p = 0.045). The physical functioning domain was associated with income of up to 1 minimum wage (p = 0.037).

## Discussion

The main finding in our study was the association between socioeconomic profile, the presence of depressive symptoms and aspects of the participants QoL. When the symptoms were present, the aspects of QoL that stood out with the highest scores were the mental health domain in Brazil and the general health perceptions and physical functioning in Portugal. In this context, these domains were associated mainly with woman group, lower education and lower income. Considering the socioeconomic analysis in the presence of depressive symptoms, the group from Brazil had higher QoL scores than the group from Portugal.

These findings may contribute to the situational diagnosis of the scenarios studied, in addition to filling scientific gaps on the subject and proposing the formulation of new social and mental health policies in the context of preventing depressive symptoms and promoting active aging and better QoL among older individuals.

In the initial analysis of the socioeconomic profile, we observed in our sample that both countries had more older women and younger people, similar to data found in other studies [17–19]. In this context, a study conducted in the United Kingdom with data extracted from the "UK Biobank" identified the woman and younger age groups as important risk factors for the development of chronic depression and at higher levels [20]. However, the highlight in our initial analysis was the predominance of lower education level and lower family income.

**Table 4. Association between quality of life (SF-36) and depressive symptoms (GDS-15) in Brazil and Portugal.**

| QoL (SF-36) | Depressive Symptoms (GDS-15) | | | |
|---|---|---|---|---|
| | Brazil (*n* = 100) | | Portugal (*n* = 50) | |
| | Absent (*n* = 41) | Present (*n* = 59) | Absent (*n* = 31) | Present (*n* = 19) |
| | Percentile | | Percentile | |
| | 25-50-75 | 25-50-75 | 25-50-75 | 25-50-75 |
| **Domains** | | | | |
| Emotional role functioning | 50-100-100 | 0-100-100 | 100-100-100 | 0-100-100 |
| p [a] | 0.071 | | 0.007 | |
| Physical functioning | 12-100-100 | 0-25-100 | 100-100-100 | 25-100-100 |
| p | 0.016 | | 0.013 | |
| Physical role functioning | 57-80-90 | 20-50-90 | 70-85-100 | 20-50-75 |
| p | 0.004 | | <0.001 | |
| Vitality | 45-50-60 | 45-50-60 | 45-55-60 | 40-45-50 |
| p | 0.818 | | 0.013 | |
| General health perceptions | 30-35-50 | 30-40-60 | 30-45-55 | 40-50-65 |
| p | 0.198 | | 0.052 | |
| Mental Health | 56-60-60 | 52-56-60 | 52-60-64 | 40-48-56 |
| p | 0.185 | | 0.005 | |
| Pain | 20-40-50 | 20-40-70 | 10-30-30 | 30-40-60 |
| p | 0.429 | | 0.002 | |
| Social role functioning | 50-50-50 | 50-50-62 | 50-50-50 | 37-50-50 |
| p | 0.477 | | 0.049 | |
| Total Score | 53-60-63 | 40-53-59 | 57-61-63 | 37-54-65 |
| p | 0.002 | | 0.049 | |
| **Dimensions** | | | | |
| Mental Health | 50-58-61 | 43-56-61 | 56-59-62 | 40-50-61 |
| p | 0.216 | | 0.022 | |
| Physical health | 47-54-59 | 38-45-56 | 56-59-63 | 43-52-58 |
| p | 0.006 | | 0.011 | |

[a] Nonparametric Mann-Whitney U test (p value)

Regarding education and income, it is known that these aspects are determinant factors for QoL [21] and for the onset or worsening of mental disorders in the general population [22]. In our sample, depressive symptoms were more common among brazilians, which is similar to that found in other comparative studies conducted with both countries, which showed better QoL among portuguese individuals [4, 6].

The descriptive analysis of QoL in our study revealed that the physical functioning, social role functioning, general health perceptions domains and physical health dimension presented the highest scores. In the comparison between groups, Portugal exhibited better performance in these aspects. A previous study showed that older portuguese people have good evaluations regarding the preservation of physical capacity and its association with QoL, cognitive and functional aspects [19]. Although we did not specifically study these associations in our study, it is important to consider that they may have influenced the results, especially among the portuguese.

The group from Portugal also stood out when we analyzed the association between QoL and the variables of depressive symptoms (absent/present). The group from Brazil showed an association between the absence of symptoms and better scores in the physical functioning

**Table 5. Multivariate analysis of socioeconomic aspects and quality of life (SF-36) in the presence of depressive symptoms (GDS-15) in Brazil and Portugal.**

| Socioeconomic Aspects | Presence of Depressive Symptoms (n = 78) | Brazil (n = 59) | Portugal (n = 19) | p [a] |
|---|---|---|---|---|
| | Domains QoL (SF-36) | Percentile 25-50-75 | Percentile 25-50-75 | |
| Gender (Woman) (n = 63) | General health perceptions | 30-40-65 | 45-57-68 | 0.042 |
| | Mental health | 48-56-60 | 37-48-52 | 0.027 |
| | Social role functioning | 50-50-62 | 28-43-50 | 0.022 |
| Age range (65–80 years) (n = 68) | Vitality | 40-50-60 | 40-45-62 | 0.032 |
| | Mental health | 51-56-61 | 39-48-54 | 0.042 |
| | Social role functioning | 38-50-62 | 34-43-50 | 0.018 |
| Marital Status (Single/widowed/divorced) (n = 37) | Mental health | 52-56-62 | 41-48-51 | 0.029 |
| Educational entertainment (≤ 5 years) (n = 64) | General health perceptions | 30-35-60 | 42-60-67 | 0.045 |
| | Mental health | 52-56-64 | 38-48-54 | 0.011 |
| Live alone (Yes) (n = 15) | Vitality | 48-55-85 | 30-40-47 | 0.012 |
| Household income (≤ 1 minimum wage) (n = 46) | Physical functioning | 0-25-75 | 25-100-100 | 0.030 |
| | Mental health | 52-56-60 | 40-48-56 | 0.037 |

[a] Nonparametric Mann-Whitney U test (p value)

and physical role functioning domains and the Physical health dimension. Portugal presented the same behavior for all domains and aspects except the general health perceptions, which did not present any significance. Thus, in addition to the physical functioning, physical role functioning and emotional role functioning, vitality and social role functioning are elements that contribute to higher levels of QoL.

Vitality is measured in the SF-36 as the level of energy, vigor and exhaustion perceived by the individual throughout the day. The social role functioning is defined as the role that the person plays within their family life and with other people in their community and is measured with a focus on their level of interaction and participation [5]. In this regard, interventions focused on social interaction and group activities have already demonstrated success in improving QoL in older people in PHC centers [23], and perceived social support exerts a protective effect against mental disorders in older people and those with multimorbidities [24]. An italian study showed a direct relationship between improvement in social function and vitality and reduction in the presence of pain [25], which was also assessed in our study.

However, pain was apparently controversial in our results because its lower QoL scores were significant in people without depressive symptoms in the portuguese group. Nevertheless, we did not observe high levels of QoL related to pain. It should be noted that pain is one of the main complaints among older people, and it has a marked impact on their QoL. Therefore, it presents itself as a limiting factor for the performance of daily activities and may negatively influence social life, mental health and QoL [25].

The multivariate analysis of the association between QoL and socioeconomic variables among participants with depressive symptoms showed that the mental health domain was the only domain that showed a significant association with all the socioeconomic variables analyzed. In all of them, we observed that the group from Brazil exhibited better scores. In this regard, there is a strong influence of mental aspects on the presence of depressive symptoms, as shown in the literature [26], and social conditions such as gender (woman), low family income and low level of education are risk factors for mental health impairment [17].

Another domain that was relevant in relation to woman group and the younger age group (65 to 80 years) was social role functioning. The participants from Brazil also exhibited better evaluations in this regard when compared to the portuguese. These data are similar to other

studies that indicated a strong influence of gender and younger age on better social functioning [27, 28]. This findings observed among women can be explained by the greater self-care and participation of women in PHC services and their health promotion activities [29]. Many of these activities involve stimulating social interaction, such as group meetings, which require individuals to travel to the PHC units [23].

Travel hinders the presence of these people as they get older due to physical and functional limitations, which imply the need for and dependence on support from caregivers and family members [30]. These limitations may explain the better performance in social role functioning among the younger participants in our sample. However, we did not study this relationship.

The general health perceptions domain was also larger between women, as well as with lower education (up to 5 years). In this regard, a better performance was observed in the group from Portugal compared to Brazil. It is important to note that this domain is described as an individual's perception of his or her health status and how it has changed over time [5]. In addition, our data reinforce the idea of woman predominance among the older in PHC [29]. We observed that individuals with less education and with depressive symptoms had higher scores in General health perceptions. This result seems to contrast with what is found in the literature, which indicates that the lower the educational level is, the worse the QoL [21].

The vitality domain, in turn, was associated with the lowest age group (65–80 years). However, of greater interest is its association with the variable "live alone", for which the highest scores were found in Brazil, and these scores did not seem to have a negative impact on this aspect of QoL, even in the presence of depressive symptoms. This finding is similar to a study conducted in Myanmar, which found a strong association between sharing a home with other people and the absence of depression [18].

*Scherrer Junior et al. (2020)* explain that the fact that an individual lives alone is an important risk factor for suicidal ideation and is strongly related to the stress generated by physical and functional limitations for activities of daily living, especially when there is no social network support, such as caregivers or family members who live near [31]. When considering the concept of vitality presented above, it is understood that younger age groups and the presence of companions in their home environment as social support may constitute determining factors for better QoL [21].

The physical functioning domain was associated with the lowest income (up to 1 minimum wage). Even with depression present, the group from Portugal exhibited high scores in this regard, unlike the Brazilians. There are also data in the literature on the influence of income and physical health on depression and QoL [32]. From another point of view, *De Albuquerque Araujo (2022)* [33] notes that income can determine the pattern and quality of food consumption and nutritional status, which, in turn, directly affects the physical conditioning of the individual [34]. In older individuals, physical fitness is crucial for active and successful aging, as well as for the reduction of depressive symptoms [35].

Given the performances observed in Brazil and Portugal, it is important to consider that the political and cultural factors specific to each region may be determinants of QoL and depressive symptoms, especially among older people [18]. Studies have shown that depression can have different indices depending on where the individual lives, such as urban vs. rural areas [36, 37].

In addition to the socioeconomic variables in the present study, it is important to note that Brazilian PHC is historically insufficient in much of the country regarding its specialized network structure for mental health [8]. The brazilian culture dictates that services are centered on consultations with a psychiatrist and little support from other professional areas. This was demonstrated by a brazilian study conducted in four large centers, where more than 90.0% of participants with mental disorders were seen by a psychiatrist. However, when analyzing

follow-up by psychologists, nurses, social workers or occupational therapists, some centers had very low percentages [38].

In the government context, the Oswaldo Cruz Foundation presented data in 2020 showing that investments in mental health policies in Brazil have not progressed since 2016. In that year, these expenses represented less than 3.0% of investments in health, while the demand for mental health services has grown exponentially [39].

The portuguese have positively evaluated their PHC services, especially in terms of accessibility and user satisfaction [40]. Even so, there is also an important gap in meeting the demand for services related to functionality, treatment of chronic diseases and mental health care, specifically cognition deficit, anxiety and depression [30].

In 2008, the National Mental Health Plan of Portugal was proposed and enacted. The central aspect of this document was the reform of mental health services and greater investments to meet this demand. However, its implementation was interrupted by the Economic and Financial Assistance Program (2011–2014), and the recovery of this delay is classified as urgent [41].

In this context, the Comprehensive Mental Health Action Plan 2013–2030, launched by the WHO, proposes minimum goals to be achieved by 2030 and shows that countries around the world are still far from achieving adequate mental health care. The plan also emphasizes the groups at greatest risk of mental disorders, such as older individuals. In addition, it draws attention to the social and cultural differences that determine the potential impact of these risks on QoL [3], as discussed in our study.

## Study limitations

As limitations of this study, there is the limited sample, especially in the group from Portugal, where we did not reach the number of participants proposed in the sample size calculation. The cross-sectional approach does not exercise the power of inference and generalization of the results or establishment of a cause-and-effect relationship. The participants were also not asked about any complementary treatments already performed, which could add a more in-depth analysis and elucidate behaviors observed in the sample.

To minimize the limitations, a rigorous methodological approach was adopted, with the pairing between the groups from Brazil and Portugal, which allowed a greater similarity between the profiles of both to reduce interference of other aspects in the variables of interest (depressive symptoms and QoL).

## Conclusions

The results revealed the existence of an association between socioeconomic status and QoL in the presence of depressive symptoms. This association was observed mainly among woman and participants with a low level of education and low income with QoL aspects related to perceived mental, physical and social health. Comparatively, the group from Brazil had higher QoL scores than the group from Portugal, and the data confirmed the hypothesis of the study.

Our conclusions point to the need for further research between the two countries regarding socioeconomic status, QoL and depressive symptoms, especially regarding the inclusion of other study variables that potentially influence the QoL of older individuals, such as the presence of certain chronic diseases. Thus, it is imperative to propose strategies to support public policies for older individuals in the field of mental health and to strengthen actions and health promotions in this population to minimize clinical complications and promote active and healthy aging.

## Author Contributions

**Conceptualization:** Larissa Silva Sadovski Torres, Mayara Priscilla Dantas Araújo.

**Data curation:** Gilson de Vasconcelos Torres.

**Formal analysis:** Gilson de Vasconcelos Torres.

**Funding acquisition:** Gilson de Vasconcelos Torres.

**Investigation:** Larissa Silva Sadovski Torres, Maria Débora Silva de Carvalho, Lívia Batista da Silva Fernandes Barbosa.

**Methodology:** Larissa Silva Sadovski Torres, Mayara Priscilla Dantas Araújo, Maria Débora Silva de Carvalho, Lívia Batista da Silva Fernandes Barbosa, Bruno Araújo da Silva Dantas, Francisco Arnoldo Nunes de Miranda, Felismina Rosa Parreira Mendes.

**Project administration:** Carmelo Sergio Gómez Martínez, Felismina Rosa Parreira Mendes, Gilson de Vasconcelos Torres.

**Resources:** Mayara Priscilla Dantas Araújo, Maria Débora Silva de Carvalho, Lívia Batista da Silva Fernandes Barbosa.

**Supervision:** Adriana Catarina de Souza Oliveira, Bruno Araújo da Silva Dantas, Carmelo Sergio Gómez Martínez, Francisco Arnoldo Nunes de Miranda, Felismina Rosa Parreira Mendes, Gilson de Vasconcelos Torres.

**Validation:** Adriana Catarina de Souza Oliveira, Bruno Araújo da Silva Dantas, Carmelo Sergio Gómez Martínez, Francisco Arnoldo Nunes de Miranda, Felismina Rosa Parreira Mendes, Gilson de Vasconcelos Torres.

**Visualization:** Francisco Arnoldo Nunes de Miranda, Gilson de Vasconcelos Torres.

**Writing – original draft:** Larissa Silva Sadovski Torres, Mayara Priscilla Dantas Araújo.

**Writing – review & editing:** Adriana Catarina de Souza Oliveira, Bruno Araújo da Silva Dantas, Carmelo Sergio Gómez Martínez, Francisco Arnoldo Nunes de Miranda, Felismina Rosa Parreira Mendes, Gilson de Vasconcelos Torres.

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
