## [Decision Letter · Decision Letter 0]

11 Apr 2023

PONE-D-22-34936Determinants of socioeconomic factors for quality of life and depressive symptoms in community-dwelling older people: a cross-sectional study in Brazil and Portugal

PLOS ONE

Dear Dr. Dantas,

Thank you for submitting your manuscript to PLOS ONE. After careful consideration, we feel that it has merit but does not fully meet PLOS ONE’s publication criteria as it currently stands. Therefore, we invite you to submit a revised version of the manuscript that addresses the points raised during the review process.

The reviewers have provided their comments for the improvement of the manuscript. They do raise some further points that I recommend are considered in any subsequent revision. These four key points that would need to be addressed can be summarized as:

1.
Review spelling, grammar, and punctuation throughout the manuscript.2.
Improve the introduction by providing more context about active aging and explaining how it relates to the overall conceptual framework of the study.3.
Use appropriate terminology when referring to the study population, such as those recommended by the World Health Organization, and avoid using biased language.4.
Check for consistency in terminology throughout the manuscript. In addition, please address the following:1.
Ensure that supporting information is auxiliary to the main content of the article. 2.
Provide a caption for any supporting information included in the manuscript, and while citation of supporting information is not strictly required, it is recommended. 3.
Clarify any differences in the hypotheses, population, and/or methods presented in this manuscript compared to those of your previously published articles on this topic. 

We look forward to receiving your revised manuscript.

Kind regards,

Delfina Fernandes Hlashwayo, M.Sc.

Academic Editor

PLOS ONE

Journal Requirements:

2. Please change "female” or "male" to "woman” or "man" as appropriate, when used as a noun (see for instance https://apastyle.apa.org/style-grammar-guidelines/bias-free-language/gender).

Reviewers' comments:

Reviewer's Responses to Questions

**Comments to the Author**

1. Is the manuscript technically sound, and do the data support the conclusions?

Reviewer #1: Yes

Reviewer #2: Yes

2. Has the statistical analysis been performed appropriately and rigorously? 

Reviewer #1: Yes

Reviewer #2: Yes

3. Have the authors made all data underlying the findings in their manuscript fully available?

Reviewer #1: Yes

Reviewer #2: Yes

4. Is the manuscript presented in an intelligible fashion and written in standard English?

Reviewer #1: Yes

Reviewer #2: Yes

5. Review Comments to the Author

Reviewer #1: The manuscript presents an important theme that contributes to the study of geriatric depression in relation to socioeconomic factors. The authors present an adequate literature review that discusses in depth all the most relevant elements. I have only small suggestions that can be changed to improve understanding of what has been presented.

General

- Throughout the text, the abbreviation for Quality of Life should be changed to “QoL” instead of “QOL”.

Results

- In tables 1, 2, 3, 4 and 5, some data is bold. The reason for this is not very clear. I suggest that authors enter this information in the legend of each table or delete this highlight.

Reviewer #2: Dear Author,

Congratulations on this well-written manuscript. My comments are as follows:

1. Line 45 (Suggestion): Change "/" from "(76.0%/p = 0.224)" to "(76.0%, p = 0.224)"

2. Line 86, 91 and 94..: Check grammar throughout manuscript. Position of full stop should be consistent with citation - etc., change "(QOL)[4]." to "(QOL).[4]"

3. Introduction: It would be helpful for the readers if the authors were to define active ageing and how it relates to the overall conceptual framework of this study

4. Line 298-300...: There are a few capitalised words which I think would be better left lower case; similar to other words in the discussion - please check

5. Line 314-317 would fit better narratively in the introduction

6. Line 327: Double check the word "elderly people", it would be more appropriate to use "older adults/people" Some examples of biased language are “elder” or “elderly”, “senior”, or “the aged”. It is advisable against using these terms because they evoke negative stereotypes of older adults, which can lead to othering older adults, bias against older adults, and poor outcomes for older adults.

Thank you.

6. PLOS authors have the option to publish the peer review history of their article (what does this mean?). If published, this will include your full peer review and any attached files.

Reviewer #1: No

Reviewer #2: **Yes: **Hussein Rizal

---

## [Author Response · Author response to Decision Letter 0]

17 Apr 2023

Dear editor,

The following are responses to each reviewer's comment:

Reviewer #1: 

The manuscript presents an important theme that contributes to the study of geriatric depression in relation to socioeconomic factors. The authors present an adequate literature review that discusses in depth all the most relevant elements. I have only small suggestions that can be changed to improve understanding of what has been presented.

General

- Throughout the text, the abbreviation for Quality of Life should be changed to “QoL” instead of “QOL”.

Answer: Corrections regarding abbreviations have been made.

Results

- In tables 1, 2, 3, 4 and 5, some data is bold. The reason for this is not very clear. I suggest that authors enter this information in the legend of each table or delete this highlight.

Answer: Regarding the bold in some values, we would like to visually highlight the results with statistical significance. However, in view of your note, we prefer to remove it to avoid confusion for readers. Thanks.

Reviewer #2: 

Dear Author,

Congratulations on this well-written manuscript. My comments are as follows:

1. Line 45 (Suggestion): Change "/" from "(76.0%/p = 0.224)" to "(76.0%, p = 0.224)"

Answer: Change made.

2. Line 86, 91 and 94..: Check grammar throughout manuscript. Position of full stop should be consistent with citation - etc., change "(QOL)[4]." to "(QOL).[4]"

Answer: The model sent by the Plos One journal advises that the period should be after the citation. However, there must be a space between the last word and the quote. We have made this change and believe this resolves this issue.

3. Introduction: It would be helpful for the readers if the authors were to define active ageing and how it relates to the overall conceptual framework of this study

Answer: We added a paragraph about that. Thanks for the suggestion.

4. Line 298-300...: There are a few capitalised words which I think would be better left lower case; similar to other words in the discussion - please check

Answer: We checked and made the necessary corrections.

5. Line 314-317 would fit better narratively in the introduction

Answer: The concepts brought in lines 314-317 refer to specific variables of the SF-36 and we thought it best to keep this text in the discussion. However, we thought it worth adding this relationship in the introduction and we did.

6. Line 327: Double check the word "elderly people", it would be more appropriate to use "older adults/people" Some examples of biased language are “elder” or “elderly”, “senior”, or “the aged”. It is advisable against using these terms because they evoke negative stereotypes of older adults, which can lead to othering older adults, bias against older adults, and poor outcomes for older adults.

Answer: We standardized the use of the term “older people” throughout the text.

Thank you.

We strive to contemplate all notes in order to further improve our work. In addition to the responses to reviewers, I would also like to send responses to the Journal Requirements, which are listed below:

Journal Requirements:

Answer: We reviewed all the material and made the necessary adjustments.

2. Please change "female” or "male" to "woman” or "man" as appropriate, when used as a noun (see for instance https://apastyle.apa.org/style-grammar-guidelines/bias-free-language/gender).

Answer: Change made.

Answer: Information added in the body of the manuscript and on the submission platform.

Answer: Sorry. We have carefully read the manuscript and the submission platform and have not seen the “Financial Disclosure” section. Please help us to find and verify this inconsistency. However, we realized that we filled in the funding grant number incorrectly and made the change.

Answer: We noticed that we filled in the funding grant number incorrectly and made the change. Thanks for the remarks.

---

## [Decision Letter · Decision Letter 1]

1 Jun 2023

Determinants of socioeconomic factors for quality of life and depressive symptoms in community-dwelling older people: A cross-sectional study in Brazil and Portugal

PONE-D-22-34936R1

Dear Dr. Dantas,

We’re pleased to inform you that your manuscript has been judged scientifically suitable for publication and will be formally accepted for publication once it meets all outstanding technical requirements.

Kind regards,

Delfina Fernandes Hlashwayo, M.Sc.

Academic Editor

PLOS ONE

Additional Editor Comments (optional):

According to the journal guidelines, it is important to ensure that any supporting information included with the manuscript serves as supplementary material to the main content of the article. I would strongly discourage the author from including the editing certificate as a supplemental file. Manuscript submissions typically do not require or expect an editing certificate as part of the supplementary materials. Instead, I suggest the author acknowledges the editing assistance in the acknowledgments section of the manuscript.

Reviewers' comments:

Reviewer's Responses to Questions

**Comments to the Author**

1. If the authors have adequately addressed your comments raised in a previous round of review and you feel that this manuscript is now acceptable for publication, you may indicate that here to bypass the “Comments to the Author” section, enter your conflict of interest statement in the “Confidential to Editor” section, and submit your "Accept" recommendation.

Reviewer #1: All comments have been addressed

Reviewer #2: All comments have been addressed

2. Is the manuscript technically sound, and do the data support the conclusions?

Reviewer #1: Yes

Reviewer #2: Yes

3. Has the statistical analysis been performed appropriately and rigorously? 

Reviewer #1: Yes

Reviewer #2: Yes

4. Have the authors made all data underlying the findings in their manuscript fully available?

Reviewer #1: Yes

Reviewer #2: Yes

5. Is the manuscript presented in an intelligible fashion and written in standard English?

Reviewer #1: Yes

Reviewer #2: Yes

6. Review Comments to the Author

Reviewer #1: (No Response)

Reviewer #2: (No Response)

7. PLOS authors have the option to publish the peer review history of their article (what does this mean?). If published, this will include your full peer review and any attached files.

Reviewer #1: No

Reviewer #2: **Yes: **Hussein Rizal

---

## [Editor Report · Acceptance letter]

5 Jun 2023

PONE-D-22-34936R1 

Determinants of socioeconomic factors for quality of life and depressive symptoms in community-dwelling older people: A cross-sectional study in Brazil and Portugal 

Dear Dr. Dantas:

I'm pleased to inform you that your manuscript has been deemed suitable for publication in PLOS ONE. Congratulations! Your manuscript is now with our production department. 

Kind regards, 

on behalf of

Ms. Delfina Fernandes Hlashwayo 

Academic Editor

PLOS ONE